# An Activated Bismuth Layer Formed In Situ on a Solid Bismuth Microelectrode for Electrochemical Sensitive Determination of Ga(III)

**DOI:** 10.3390/membranes12121267

**Published:** 2022-12-15

**Authors:** Malgorzata Grabarczyk, Edyta Wlazlowska

**Affiliations:** Department of Analytical Chemistry, Institute of Chemical Sciences, Faculty of Chemistry, Maria Curie-Sklodowska University, 20-031 Lublin, Poland

**Keywords:** gallium, electrochemical measurement, stripping voltammetry, activated bismuth layer formed in situ on a solid bismuth microelectrode

## Abstract

In this paper, an activated bismuth layer formed in situ on a solid bismuth microelectrode, used as a working electrode for the electrochemical sensitive determination of Ga(III), based on anodic stripping voltammetry (ASV) is discussed. The new electrode significantly enhances the sensitivity in the ASV determination of Ga(III) and exhibits superior performance in comparison to a bismuth film electrode prepared on a glassy carbon disc. The experimental variables, such as the potential and time of solid-bismuth-microelectrode activation, the composition of the supporting electrolyte, and the influence of possible interferences on the Ga(III) signal response, were tested. The most favorable values were selected (pH = 4.6; acetate buffer; activation potential/time: −1.8 V/6 s and −1.4 V/60 s). In the optimized conditions, the peak current was found to be proportional to the concentration of Ga(III) over the range from 2 × 10^−8^ to 2 × 10^−6^ mol L^−1^ with R = 0.993. The limit of detection (LOD) was 7 × 10^−9^ mol L^−1^. Finally, the proposed method was successfully applied for gallium determination in certified reference waters, such as surface water and waste water, as well as tap and river water samples. The water samples were analyzed without any pretreatment and recovery values from 92.4 to 105.5% were obtained.

## 1. Introduction

The interest in the determination of Ga(III) has been very high for many years and is not waning. It is related to its wide application in various fields. Gallium is used in compounds such as metallic gallium, gallium antimonide, gallium arsenide, gallium chemicals, gallium nitride, and gallium phosphide. Pure gallium or gallium fused with other metals is used in many devices, such as high current switches, pressure gauges, thermometers, and many others. Gallium compounds are widely used in electronic and optoelectric devices operating in infrared. Gallium arsenide has military applications in radios, satellites, night-vision devices, and in communication equipment. In the case of gallium nitride, it is used to produce various types of laser diodes. These lamps use less energy than conventional incandescent electric lamps and do not contain any toxic heavy metals. Radioactive gallium and gallium nitrate are used in clinical medicine as diagnostic and therapeutic agents in neoplastic diseases and disorders of calcium and bone metabolism [1,2,3,4].

Such a wide use of gallium compounds is associated with an increased possibility of its release to the environment. One of main sources of gallium in the environment is gallium released in semiconductor processing. When analyzing the influence of gallium on humans, it was noticed that the majority of people working in semiconductor companies are many times more exposed to the effects of gallium compounds. Potential exposure may be encountered during the production of inorganic gallium compounds, smelting of gallium metal, recycling of electrical waste, or the use of gallium-containing agents for therapeutic purposes. The effects of poisoning with gallium compounds can manifest as a skin rash, increased heart rate, shortness of breath, dizziness, and headache. Nevertheless, gallium toxicity is relatively low compared to its anti-disease effectiveness [5,6,7,8,9]. Nevertheless, environmental monitoring is necessary to determine even trace concentrations of gallium in various types of samples, including water samples, as water facilitates the movement of pollutants in the environment.

The most widely accepted procedures used in the determination of gallium are electrochemical methods, among others, with an anodic stripping voltametric 2 mode. Anodic stripping voltammetry (ASV) is the oldest stripping method used [10]. Thanks to its high sensitivity and low limit of quantification, this method is used in the analysis of many natural and industrial samples. In the ASV method, the electrolysis process is carried out by the electrochemical reduction in a substance with a measured value at constant potential, combined with the simultaneous mixing of the solution. After electrolytic concentration, the electrode is polarized in the positive direction at an appropriate value of the potential and the released metal goes into the solution; this process involves recording the current in the form of a voltamperogram. For many years, mercury electrodes have been the main choice in the voltammetric analysis of gallium and other metal ions. However, in recent years, safety and environmental considerations have restricted their use and encouraged the search for alternative materials that are more environmentally friendly and capable of possessing more possibilities for in situ and flow analysis. These include, inter alia, bismuth electrodes, introduced in 2000 by Joseph Wang and his group as bismuth film electrodes on glassy carbon substrate for ASV measurements of heavy metals [11]. Their studies demonstrated that bismuth-coated glassy carbon electrodes offer an attractive stripping voltammetric performance. The attractive stripping behavior of bismuth electrodes reflects the ability of bismuth to form fused multicomponent alloys with heavy metals, which is used in the ASV method where the determined metal is reduced to a metallic form during the accumulation step [12]. As for the voltammetric methods of Ga(III) determination, the first works of Moorhead et al. using the anodic striping voltammetry method appeared in the 1970s [13]. Further research led to the development of another ASV procedure using a hanging mercury drop electrode (HMDE) with the differential pulse technique, reducing the detection limit [14]. In order to eliminate the HMDE as a source of metallic mercury, the mercury film electrode (MFE), plated electrochemically in situ from a solution containing Hg(II) ions and added into the supporting electrolyte solution, was proposed to be used as the working electrode [15]. Another step in the development of methods limiting the use of mercury for the determination of Ga(III) by the ASV method was proposed by Piech et al. to use a new cyclic renewable silver-amalgam film electrode. The design of this electrode drastically reduced the amount of mercury used for one measurement and limited its contact with the laboratory atmosphere [16]. The most important breakthrough enabling the complete elimination of mercury was the introduction of film bismuth electrodes and the publication concerning the Ga(III)-determination procedure using the ASV method by J.V. Kamat et al. in 2011 [17]. However, this was associated with a loss in sensitivity compared to the HMDE procedure. Therefore, the aim of our research was to use a non-toxic bismuth electrode with a simultaneous increase in the sensitivity of Ga(III) determinations. This was achieved by replacing glassy carbon as the substrate for creating bismuth films, as was conducted in paper [17], by an activated bismuth layer formed in situ on a solid bismuth microelectrode. The solid bismuth microelectrode has been described previously, and it has been successfully applied to the determination of folic acid, Tl(I), V(V), and Sn(II). In these studies, the accumulation of the analyte to be determined was carried out directly on the solid bismuth microelectrode [18,19,20]. As proven in this work, the increase in the sensitivity of Ga(III) determinations was achieved thanks to the electrochemical activated bismuth layer formed in situ on the solid bismuth microelectrode. Additionally, the strength of the developed Ga(III)-determination procedure, combining the ASV technique with the activated bismuth layer formed in situ on the solid bismuth microelectrode, offers a simple and quick way to reduce the negative impact of the matrix, which allowed for the direct analysis of real environmental samples. The negative matrix effect is most often associated with the blocking of the surface of the working electrode due to the adsorption of sample matrix components, such as surfactants or humic substances. The use of an XAD-7 resin additive to the analyzed sample causes the absorption of the aforementioned substances on the resin so that they are not adsorbed on the surface of the working electrode and do not interfere with the voltammetric measurements.

## 2. Experimental

### 2.1. Apparatus

The stripping voltammetry measurements were performed with µAutolab from ECO/Chemie (Utrecht, The Netherlands). The measurements were typically accomplished using a three-electrode system containing a working, counter, and reference electrode and a 10 mL quartz cell. The solid bismuth microelectrode (a diameter of 25 µm) was used as the working electrode, the Pt plate was used as the counter electrode, and a Ag/AgCl system was used as the reference electrode. A detailed description of the design of the solid bismuth microelectrode was presented in paper [21]. The solid bismuth microelectrode was placed in housing made of PEEK, and its real image is presented in Figure 1, which was taken with an Inverted Metallographic Microscope Nikon MA200 (Tokyo, Japan). The electrode was polished daily on 2500 grit sandpaper, then rinsed with copious amounts of triple-distilled water, and kept in an ultrasonic bath for 30 s to remove any residual polishing material. Visible irregularities on the surface of metallic bismuth (Figure 1) were related to the manual polishing of the electrode on sandpaper at the beginning of each measurement day. However, it was proven that this does not affect the stability of the measurement. Comparing the surface of the solid bismuth electrode to the surface of the BiFE electrodes, it can be concluded that in the case of a solid bismuth electrode, metallic bismuth is visible on its entire surface, while in BiFE electrodes generated on glassy carbon, bismuth coats a much smaller part of the surface compared to the surface of glassy carbon on which it is generated [22,23].

### 2.2. Reagents

All chemicals used were of analytical reagent grade or Suprapur. For all purposes, distilled water obtained from a water purification Milli-Q system (Millipore, London, UK) was used. An acetate buffer was prepared from Suprapur CH3COOH and NaOH obtained from Merck. A stock solution of 1 g L^−1^ Ga(III) was obtained from Merck (Darmstadt, Germany). The solutions of lower Ga(III) concentrations were prepared by dilution of the stock solution as required. A standard solution of 1 g L^−1^ Bi(III), Triton X-100, sodium dodecyl sulfate (SDS), cetyltrimethylammonium bromide (CTAB), and Rhamnolipid were acquired from Fluka (Buchs, Switzerland). Humic acid (HA) sodium salt was purchased from Aldrich. Natural organic matter (NOM) and river fulvic acid (FA), originating from the Suwannee River, were purchased from the International Humic Substances Society. Amberlite XAD-7, acquired from Sigma, was washed four times in distilled water and dried at a temperature of 50 °C. For the validation of the procedure, the standard materials ‘‘SPS-WW1—Reference Material for Measurement of Elements in Wastewaters’’ and “SPS-SW2—Reference Material for Measurement of Elements in Surface Waters’’ from Spectrapure Standards, Oslo, Norway, were used.

### 2.3. Real Sample Preparation

Tap water samples were taken from the laboratory. Water samples from the Bystrzyca river were collected in clean polypropylene bottles and filtered through a 0.45 µm cellulose-acetate-membrane filter in Sartorius apparatus. Because certified reference materials SPS-WW1 and SPS-SW2 contain nitric acid, for every 1 mL of their solution, 100 µL of sodium hydroxide at a concentration of 1 mol L^−1^ was added to neutralize the pH.

### 2.4. Measurement Procedure

Unless otherwise indicated, the general procedure for voltammetric measurements was carried out as follows. The analyzed samples or gallium solution, 1 mL of 1 mol L^−1^ acetate buffer (pH = 4.6), and 250 µL of 1 g L^−1^ Bi(III) were placed into the voltammetric cell and then distilled water was added at a volume of 10 mL. Each measurement consisted of activating the solid bismuth microelectrode by depositing a fresh layer of bismuth on it through the electrochemical reduction in Bi(III) ions present in the solution as a result of the successive potentials of −1.8 V for 6 s and −1.4 V for 60 s. During this step, the solution was stirred by a magnetic bar. The application of the above conditions enabled both the activation of the bismuth electrode and the accumulation of Ga(III) as a result of its reduction to a metallic form. Then, stirring was stopped, the solution was left to settle for 5 s, and voltammetric stripping was performed in the potential range from −1.2 to −0.4 V. In the signal recording stage, the oxidation of metallic gallium occurred as a result of the potential change toward less negative potentials, and due to this process, a well-formed peak was visible on the voltammogram, which is the basis for quantitative Ga(III) analysis in the proposed procedure. All measurements were executed at room temperature in the presence of oxygen.

## 3. Results and Discussion

### 3.1. Activation Conditions of the Bismuth Layer on the Solid Bismuth Microelectrode

As proven in previous works, it is possible to concentrate the analyte using the solid bismuth electrode [21,24]. In the proposed measuring system, a gallium peak was also observed after its accumulation as a result of its reduction to a metallic form, and then a signal was obtained after its oxidation. However, a very significant increase was observed in the sensitivity of the determinations due to the activation of the solid bismuth electrode by introducing Bi(III) ions into the tested solution and their simultaneous reduction with the accumulated gallium. The activation of the electrode was obtained as a result of the precipitation of freshly reduced metallic bismuth on the surface of the solid bismuth electrode and, due to that, the electrochemical activity of the electrode increased. Therefore, a series of measurements were carried out to select the optimal activation conditions of the bismuth layer on the solid bismuth microelectrode. It was found that the best results were obtained using two-stage activation with two different negative potentials applied to the solid bismuth electrode. Therefore, their influence was investigated by changing the values of the potentials and their duration in such a way that one parameter was changed, while the others were unchanged, and the gallium signal recorded under such conditions was observed. Under optimized conditions, the activation pattern was as follows: −1.8 V for 6 s and −1.4 V for 60 s. These conditions were established on the basis of the conducted experiments, the results of which are presented in Figure 2A,B. As can be seen from curve “a” in Figure 2A, the value of the first activation potential changed in the range from −2.6 V to −1.4 V, using a constant time of 6 s. It was observed that by changing the potential towards values more negative and more positive than −1.8 V, the gallium peak current decreased. By changing the value of the second activation potential in the range from −1.8 V to −1.0 V (curve “b” in Figure 2A), it was observed that by changing the potential towards values more positive than −1.4 V, the gallium peak decreased in the potential range of −1.4 V and −1.8 The V peak was of a similar size, but at more negative potentials, the peak was less well-formed. Therefore, the potentials of −1.8 V and −1.4 V were selected as the optimal potentials for electrode activation.

The next step in the research was to investigate the effect of the activation time on the solid bismuth electrode. For this purpose, the first activation potential of −1.8 V was applied to the electrode during a period ranging from 1 s to 8 s, while the second activation potential of −1.4 V was applied for 60 s. As can be seen from Figure 2B (curve “a”), the gallium peak current increased as the time changed from 1 to 6 s, and then it did not change. The same experiments were carried out for the constant duration of the first activation time, i.e., −1.8 V for 6 s, and the second activation time at −1.4 V changed in the range from 10 to 80 s. As can be seen from Figure 2B (curve “b”), the gallium peak current increased by extending the accumulation time to 60 s, and then stabilized at a constant value. On the basis of the obtained results, the following alternating potential was assumed as optimal: −1.8 V for 6 s and −1.4 V for 60 s in order to activate the solid bismuth microelectrode through the bismuth layer formed in situ.

### 3.2. Effect of Acetate Buffer and Bismuth Concentration

An important step in optimization is the selection of an appropriate supporting electrolyte. Acetate buffer and acetate acid were selected for the tests, but the signal obtained with acetate acid did not meet our expectations because it was very weak. For the tests, a 0.1 mol L^−1^ acetate buffer was used in the pH range of 3 to 6, and the buffer with a pH of 4.6 turned out to be the best, guaranteeing the highest signal of the gallium peak.

In order to activate the solid bismuth electrode through the bismuth layer formed in situ, it was necessary to introduce Bi(III) ions into the analyzed solution. As was observed, the current of the gallium peak in the voltamperogram gradually increased with the increase in the Bi(III) ion concentration range. The obtained results are presented in Figure 3. As can be seen, the gallium signal increased with an increase in the concentration of Bi(III) to 1.2 × 10^−4^ mol L^−1^, and then its height did not increase. However, at the same time, its shape deteriorated and the peak became wider and wider up to the tested concentration of 2 × 10^−4^ mol L^−1^. Simultaneously, it was noticed that the background current increased successively with the increase in the concentration of Bi(III) ions in the solution. Therefore, the concentration of Bi(III) 1.2 × 10^−4^ mol L^−1^ was chosen as the most optimal for the activation of the solid bismuth microelectrode. Figure 4 shows example voltamperograms recorded for 2 × 10^−7^ mol L^−1^ Ga(III) using a solid bismuth microelectrode as a working electrode without activation and a solid bismuth microelectrode with activation under optimized conditions.

### 3.3. Analytical Performance

A series of voltammograms used to increase Ga(III) concentration under optimized conditions were recorded, and the corresponding calibration curve was prepared. The optimized conditions were as follows: composition of the solution 0.1 mol L^−1^ acetic buffer (pH = 4.6) and 1.2 × 10^−4^ L^−1^ Bi(III) in triple-distilled water; and voltammetric measurement −1.8 V for 6 s and −1.4 V for 60 s. The calibration plot was linear in the range from 2 × 10^−8^ mol L^−1^ (1.39 µg L^−1^) to 2 × 10^−6^ mol L^−1^ (139 µg L^−1^) and obeyed the equation y = 376.7x + 4.0, where y is the peak current (nA) and x is the Ga(III) concentration (µmol L^−1^). The linear correlation coefficient was r = 0.993. The calibration curve is shown in Figure 5. For the developed procedure, the limit of detection (LOD) was found to be 7 × 10^−9^ mol L^−1^ (0.48 µg L^−1^), estimated from three times the standard deviation for the lowest studied Ga(III) concentration. Comparing the obtained parameters with the parameters previously published for ASV Ga(III) determination using a classical bismuth film electrode generated on a glassy carbon electrode (linearity range from 20 to 100 µg L^−1^, LOD equal to 6.6 µg L^−1^), a significant improvement in performance was obtained as the detection limit was lower by more than an order of magnitude. Additionally, the linearity range was significantly extended [17].

### 3.4. Stability of Measurement

The short-term repeatability of the analytical response for the activated bismuth layer formed in situ on the solid bismuth microelectrode used for gallium determination was examined. The signals obtained for six different samples with the supporting electrolyte containing 2 × 10^−7^ mol L^−1^ of Ga(III) were compared, and the calculated RSD was equal to 3.1%.

The long-term stability (six months) was also tested by examining the reproducibility of the height for 2 × 10^−7^ mol L^−1^ of Ga(III). Six identical measurements were made every month, and the RSD was calculated based on the size of the obtained peaks, which was 5.5%.

A prerequisite for achieving such a high reproducibility of results has always been the same preparation of the electrode by polishing it before the start of measurements.

The stability of the electrode measurements was also tested by determining their robustness. The robustness was examined when the parameters, such as the pH and buffer concentrations, were slightly changed. We found that the obtained results did not change when those parameters were modified.

### 3.5. Tolerance to Interfering Species

Considering that other metals may also undergo a reduction in the negative potentials used in the proposed procedure and accumulate on the surface of the electrode, their influence on the gallium signal was investigated. The effects of foreign ions were tested using a fixed concentration of Ga(III) equal to 2 × 10^−7^ mol L^−1^ and different amounts of examined ions. This research showed that Al(III), Cu(II), Hg(II), Sn(II), Pb(II), V(V), and Ti(IV) at 1 × 10^−5^ mol L^−1^, as well as Cd(II), and Mo(VI) at 5 × 10^−6^ mol L^−1^, did not interfere with the gallium signal (the disturbance threshold was set to ±5% variation in the peak current). Fe(II) and Zn(II) proved to be the most interfering ions because their permissible non-interference concentration was 5 × 10^−7^ mol L^−1^.

The presence of organic substances had a huge impact on the signals obtained during the analysis of real samples in stripping voltammetry. The effect of co-existing, surface-active substances and humic substances on the stripping peak current of Ga(III) was therefore investigated.

Several surface-active substances were selected as representatives of anionic, cationic, and nonionic surfactants, i.e., SDS, CTAB, Triton X-100, and biosurfactant rhamnolipid. Humic acid (HA) and fulvic acid (FA) were selected as the representatives of humic substances, and natural organic matter (NOM) was additionally used in the tests. It turned out that humic substances, such as HA, FA, and NOM, did not affect the gallium signal at a concentration of even 5 ppm. As for the surfactants, the gallium signal was unaffected by the presence of 5 ppm SDS and rhamnolipid. An interfering effect was observed in the presence of Triton and CTAB, which is presented graphically in Figure 6.

A good and very effective method of effectively reducing these interferences is the use of the preliminary mixing of the analyzed sample with XAD-7 resin [25,26]. This has already been applied in previously described procedures for the determination of other elements. For these adsorptive stripping voltammetric procedures, an additional step of mixing with the resin had to be introduced, followed by the introduction of the sample into the voltammetric cell. Because the analyte to be determined was in a complexed form with complexones, such as DTPA, cupferron, or chloranilic acid, when the resin was introduced directly into the vessel, the analyte to be determined was also adsorbed on the resin. Therefore, it was necessary to introduce an additional mixing step with the resin before the actual measurement. In the ASV procedure described in this paper, the Ga(III) ions in the analyzed sample were in an uncomplexed form and did not adsorb to the resin. As a result, resin can be added directly to a voltammetric cell, which does not require the introduction of an additional stage before the measurement. Thus, 0.3 g of XAD-7 resin was introduced into the analyzed solutions containing the above-mentioned organic substances and the sample was mixed for 3 min. Then, the measurement was carried out in the same solution. As a result, the interfering effect of Triton X-100 and CTAB was reduced, and an undisturbed gallium signal was recorded from the solutions containing even 5 ppm of their concentrations.

### 3.6. Analytical Applications

The developed method was assessed by analysis of tap water, Bystrzyca river water, and certified reference materials, such as SPS-SW2 surface water and SPS-WW1 waste water. In all cases, a recovery test was performed after introducing appropriate amounts of gallium into the analyzed samples. This was due to the lack of appropriate reference materials containing gallium and the fact that water samples collected in the vicinity of the Lublin area did not show content of this element. Therefore, the recoveries from materials with various environmental matrices were tested in order to verify the correctness and practical use of the developed procedure. The obtained results are presented in Table 1, while the sample voltammograms recorded during the SPS-SW2 surface water analysis are shown in Figure 7. As can be seen, the obtained recovery results for all samples with a different matrix were satisfactory, which offers a great chance for the practical use of the developed procedure for monitoring a variety of real waters. For comparison, the samples were analyzed after their previous high-pressure mineralization with the use of nitric acid. The obtained results of Ga(III) recovery ranged from 94.7 to 103.4%, which confirms that the proposed procedure for the determination of Ga(III) can be successfully applied to the analysis of samples with an organic matrix, as well as samples from which this matrix was removed.

## 4. Conclusions

For a long time, since the invention of polarography heavy metals and the electrochemical analysis of some metalloids mostly with mercury electrodes, only a minority of authors had proposed modifications of solid or carbon paste electrodes as an alternative to the hegemony of mercury. However, when safety and environmental considerations advised against mercury in the new century, intensive research started to find substitutes for such a valuable but toxic material [27]. The research presented in this paper is also devoted to this issue. The novelty of this work was the first-time use of a solid bismuth microelectrode for the determination of trace amounts of Ga(III) by voltammetry. Based on the experiments that have been discussed in the previous sections, the following general remarks and conclusions can be made:-the solid bismuth microelectrode can be a valuable alternative for mercury electrodes for Ga(III) determination by anodic stripping voltammetry;-thanks to the use of an activated bismuth layer formed in situ on the solid bismuth microelectrode, the sensitivity of the determinations significantly increased compared to the solid bismuth electrode;-regarding long-term durability, after a minimum of six months of using this sensor, the voltammetric response of gallium remained practically unchanged;-the developed procedure for the determination of gallium is simple, fast, and does not require expensive equipment;-as proven, it is possible to directly analyze water environmental samples thanks to the minimizing of the negative effects of the organic matrix.

## Figures and Tables

**Figure 1 membranes-12-01267-f001:**
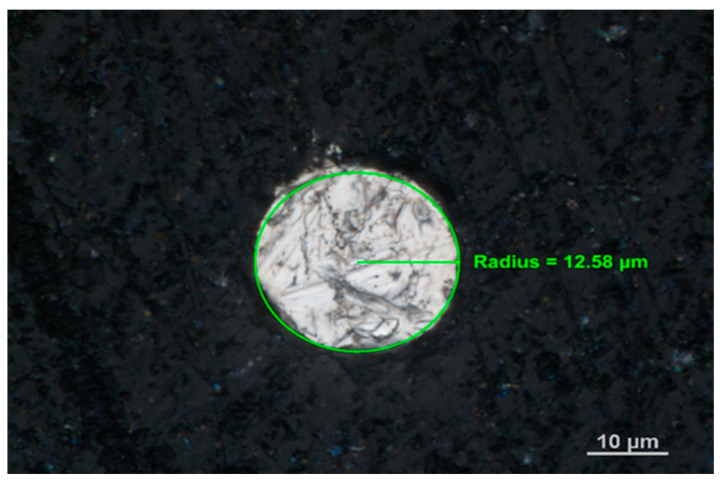
A microscopic image of the solid bismuth microelectrode with the marked radius.

**Figure 2 membranes-12-01267-f002:**
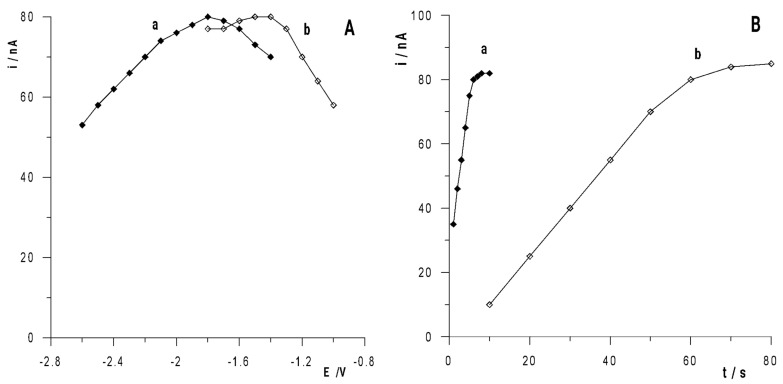
(**A**) The influence of activation potential on 2 × 10^−7^ mol L^−1^ Ga(III) peak current: (a) first activation potential was changed for 6 s, second activation potential was constant −1.4 V for 60 s; (b) first activation potential was constant −1.8 V for 6 s, second activation potential was changed for 60 s. (**B**) The influence of activation time on 2 × 10^−7^ mol L^−1^ Ga(III) peak current: (a) first activation time was changed at −1.8 V, second activation time was constant 60 s at −1.4 V; (b) first activation time was constant 6 s at −1.8 V, second activation time was changed at −1.4 V.

**Figure 3 membranes-12-01267-f003:**
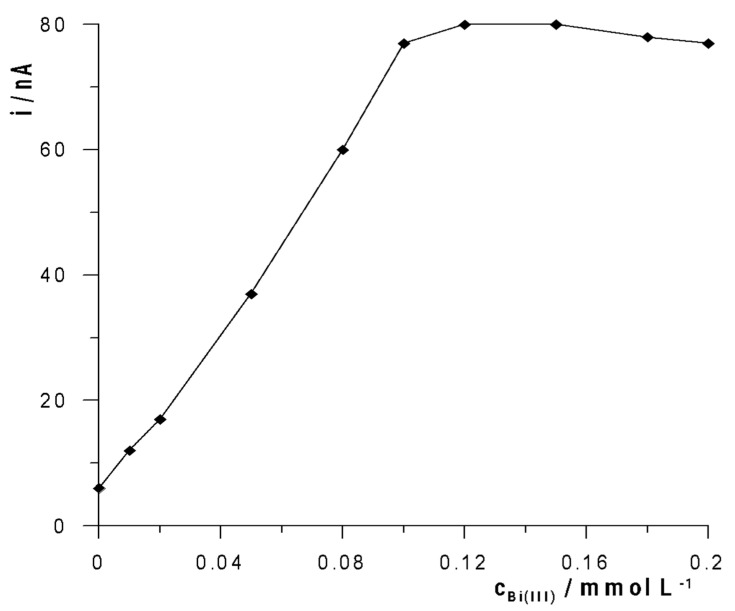
The influence of Bi(III) concentration on the 2 × 10^−7^ mol L^−1^ Ga(III) peak current.

**Figure 4 membranes-12-01267-f004:**
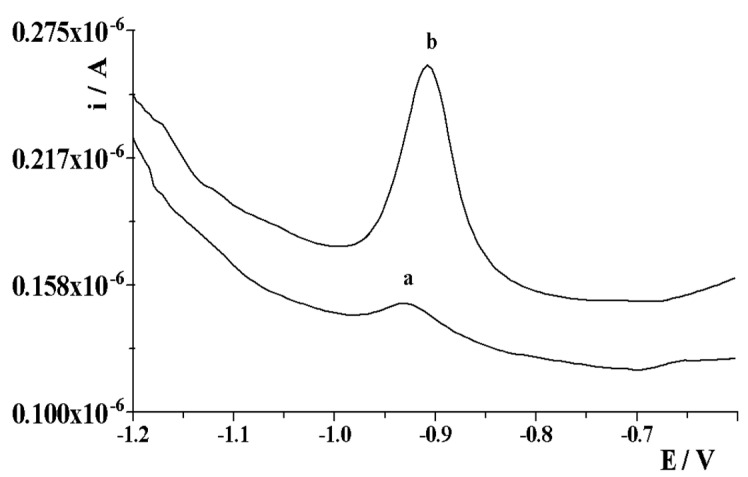
Exemplary voltamperograms recorded for 2 × 10^−7^ mol L^−1^ Ga(III) using as working electrode: solid bismuth microelectrode without activation (a); solid bismuth microelectrode with activation (b).

**Figure 5 membranes-12-01267-f005:**
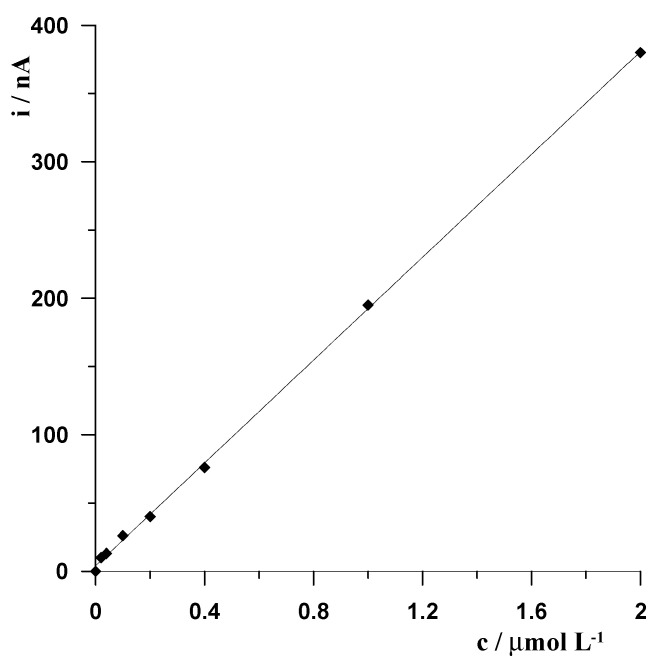
The calibration curve for the solution 0.1 mol L^−1^ acetic buffer (pH = 4.6), 1.2 × 10^−4^ L^−1^ Bi(III), and varying concentrations of Ga(III) in the range from 2 × 10^−8^ mol L^−1^ to 2 × 10^−6^ mol L^−1^.

**Figure 6 membranes-12-01267-f006:**
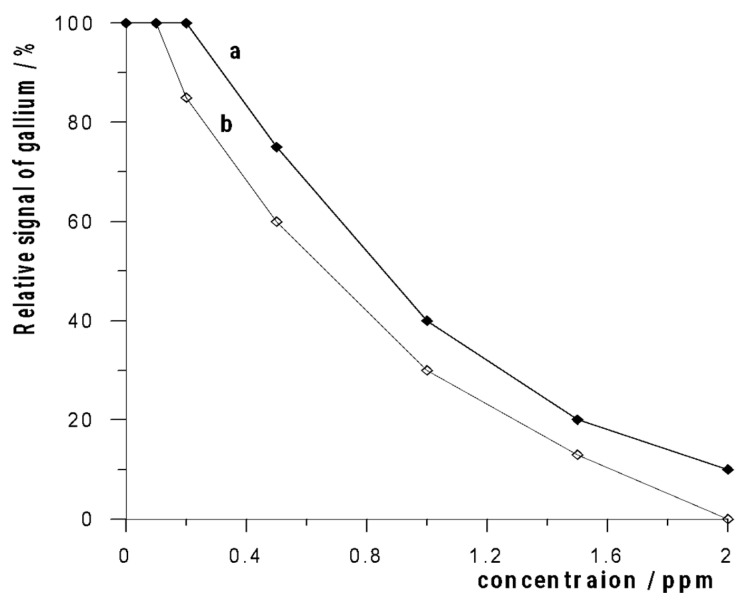
The influence of surface-active substances: Triton X-100 (a) and CTAB (b) on relative signal of 5 × 10^−7^ mol L^−1^ Ga(III).

**Figure 7 membranes-12-01267-f007:**
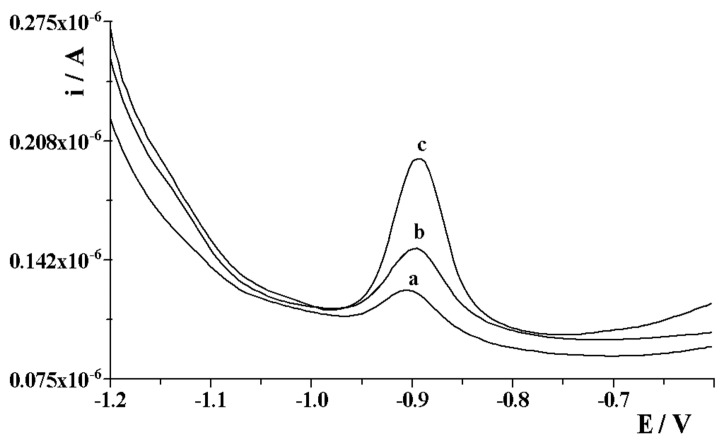
Voltammograms obtained in the course of gallium determination in Bystrzyca river water spiked with 2 × 10^−7^ mol L^−1^ Ga(III) three hours before the measurement: (a) diluted four times; (b) as (a) + 5 × 10^−8^ mol L^−1^ Ga(III); (c) as (a) + 1.5 × 10^−7^ mol L^−1^ Ga(III).

**Table 1 membranes-12-01267-t001:** Analytical results of Ga(III) determination in certified reference materials and natural water samples. The samples were analyzed after tenfold dilutions using the standard addition method.

Sample	Ga(III) Added (nmol L^−1^)	Ga(III) Found (nmol L^−1^)	Recovery (%)	RSD (n = 5) (%)
Certified reference material SPS-WW1 waste water	100	92.4	92.4	6.2
	400	382.4	95.6	4.6
Certified reference material SPS-SW2 surface water	50	52.0	104.0	3.2
	200	211	105.5	4.8
Bystrzyca river water	100	94.4	94.4	5.8
	400	384.8	96.2	5.3
Tap water	50	51.5	103.0	6.4
	200	195	97.5	4.6

## Data Availability

Not applicable.

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
