# Peer review of "An Activated Bismuth Layer Formed In Situ on a Solid Bismuth Microelectrode for Electrochemical Sensitive Determination of Ga(III)"

_membranes, 2022, doi:10.3390/membranes12121267_

Round 1
Reviewer 1 Report
This work describes an activated in situ formed bismuth layer on a solid bismuth microelectrode used as a working electrode for electrochemical sensitive determination of Ga(III) based on anodic stripping voltammetry (ASV) is discussed. The new electrode significantly enhances the sensitivity for the ASV determination of Ga(III) and exhibits superior performance in comparison to a bismuth film electrode prepared on a glassy carbon disc. The experimental variables were tested such as potential and time of solid bismuth microelectrode activation, composition of the supporting electrolyte, and influence of possible interferences on the Ga(III) signal response. The most favorable values were selected (pH = 4.6, acetic buffer, activation potential/time: -1.8 V/6 s and -1.4 V/60 s). In the optimized conditions, the peak current was found to be proportional to the concentration of Ga(III) over the range from 2 × 10-8 to 2 × 10-6 mol L-1 with R = 0.993. The limit of detection (LOD) was 7 × 10-9 mol L-1. Finally, the proposed method was successfully applied for gallium determination in certified reference waters, such as Surface Water and wastewater, as well as tap and river water samples. The water samples were analyzed without any pretreatment and recovery values from 92.4 to 105.5% were obtained.
In my opinion the determination of gallium in aqueous matrices is an interesting topic and there are few reported works, however the present article presents problems of wording, fundamentals, and problems with the English language, which should be revised. My recommendation is to reject the present article.
In the Abstract:
1.- The text should improve its redaction and English
2.- “The new electrode significantly enhances the sensitivity for the ASV determination of Ga(III) and exhibits superior performance in comparison to a bismuth film electrode prepared on a glassy carbon disc”. However, the present work does not present any experimental results to support this statement. It would be advisable to carry out studies with BiFE and compare results.
3.- What is meant by the negative impact of the matrix effect ?
4.- Figure 1 requires further analysis and interpretation. You should compare the proposed electrode with the BiFE
5.- It must be substantiated what is meant by activated electrode? How is it activated?
6.- Why does a solid bismuth electrode modified with a bismuth film have a better current response than the solid electrode (BiE) or BiFE.
7.- It should be justified why in the preconcentration stap they use two potentials (-1.8 and -1.4 V). What happens to the discharge of hydrogen if acetic /acetate buffer is used 4.6.?
8.- Although the work performs and explains the effect of organic matter, it is necessary to carry out a study of interfering cations such as Fe, Cu, Zn, Cd, among others.
9.- The role of the XAD resin in the determination of Ga should be explained
10.- It is mentioned in the text that gallium would be complexed and this statement should be substantiated.
11.- It would be advisable to carry out comparative studies with samples digested by. Microwave, UV-Vis or acid that allow evaluating the effect of organic matter in the real sample.
Author Response
The revisions which we made are as follows:
Reviewer 1:
Remark: 1. The text should improve its redaction and English.
Answer: Text and English have been improved.
Remark: 2. “The new electrode significantly enhances the sensitivity for the ASV determination of Ga(III) and exhibits superior performance in comparison to a bismuth film electrode prepared on a glassy carbon disc”. However, the present work does not present any experimental results to support this statement. It would be advisable to carry out studies with BiFE and compare results.
Answer:
Such studies have been carried out and described in details in paper: “Scope of detection and determination of gallium(III) in industrial ground water by square wave anodic stripping voltammetry on bismuth film electrode.” Talanta 86 (2011) 256-265. The work is cited in our article as [17]. In Chapter 3.3. Analytical performance a comparison of our results with the results obtained at BiFE prepared on a glassy carbon disc was added.
Remark: 3. What is meant by the negative impact of the matrix effect ?
Answer: The negative impact of the matrix effect is explained and described in Chapter 1. Introduction.
Remark: 4. Figure 1 requires further analysis and interpretation. You should compare the proposed electrode with the BiFE.
Answer: Analysis and interpretation of Fig. 1 and comparison with BiFE are described in Chapter 2.1. Apparatus.
Remark: 5. It must be substantiated what is meant by activated electrode? How is it activated?
Answer: In Chapter 3.1. Activation conditions of the bismuth layer on the solid bismuth microelectrode it has been explained what electrode activation is and how it is carried out.
Remark: 6. Why does a solid bismuth electrode modified with a bismuth film have a better current response than the solid electrode (BiE) or BiFE.
Answer: It was assumed that this is due to the fact that the electrochemical activity of the electrode increases if newly formed metallic bismuth is formed on the surface of metallic bismuth as described in Chapter 3.1. Activation conditions of the bismuth layer on the solid bismuth microelectrode.
Remark: 7. It should be justified why in the preconcentration stap they use two potentials (-1.8 and -1.4 V). What happens to the discharge of hydrogen if acetic /acetate buffer is used 4.6.?
Answer: The accumulation was carried out at two different negative potentials due to the fact that this arrangement gave the highest gallium peak. When using accumulation at only one potential value, the gallium peak was lower as shown in Figure 2A and Fig. 2B. The amount of released hydrogen is so small that it does not interfere with the determination.
Remark: 8. Although the work performs and explains the effect of organic matter, it is necessary to carry out a study of interfering cations such as Fe, Cu, Zn, Cd, among others.
Answer: The influence of foreign ions has been investigated and described in section 3.5. Tolerance to interfering species.
Remark: 9. The role of the XAD resin in the determination of Ga should be explained.
Answer: The role of XAD resin in the development of the Ga(III) determination procedure is explained at the end in Chapter 1. Introduction.
Remark: 10. It is mentioned in the text that gallium would be complexed and this statement should be substantiated.
Answer: The word “complex” in chapter 3.6. Analytical applications has been removed. The additional information about complexones was added in Chapter 3.5.
Remark: 11. It would be advisable to carry out comparative studies with samples digested by. Microwave, UV-Vis or acid that allow evaluating the effect of organic matter in the real sample.
Answer: Additional measurements were performed to determine the recovery of Ga(III) for samples previously subjected to high-pressure mineralization with nitric acid, as described in Chapter 3.6. Analytical applications.
Yours sincerely,
Prof. Malgorzata Grabarczyk
Reviewer 2 Report
The work reported in this manuscript is interesting and well-presented. But it needs some corrections and improvements before acceptance.
1- please describe the novelty of your work.
2- Considering that metals such as lead, zinc, cadmium, and copper can accumulate on the surface of the electrode in negative potentials, the author should investigate their interfering effect.
3- Please change the acetic buffer to acetate buffer in the manuscript.
4- Please use the same unit for current in the related figures (µA or nA).
5- The author should display the calibration curve in the manuscript.
6- The author should provide the related equation for calculating the LOD.
7- The Stability of measurement section should come after Analytical performance section.
Author Response
The revisions which we made are as follows:
Reviewer 2:
Remark: 1. Please describe the novelty of your work.
Answer: The novelty of the work is described in Chapter 4. Conclusions.
Remark: 2. Considering that metals such as lead, zinc, cadmium, and copper can accumulate on the surface of the electrode in negative potentials, the author should investigate their interfering effect.
Answer: The influence of foreign ions has been investigated and described in section 3.5. Tolerance to interfering species.
Remark: 3. Please change the acetic buffer to acetate buffer in the manuscript.
Answer: Throughout the paper acetic buffer was changed to acetate buffer.
Remark: 4. Please use the same unit for current in the related figures (µA or nA).
Answer: In Figure 2A, 2B and 3 the current is shown in nA, in Figure 4 and 6 the current is in A but we have no way to change this scale to nA because these are the original voltammograms recorded by the Autalab analyzer, these are the factory settings and there is no way to change the unit.
Remark: 5. The author should display the calibration curve in the manuscript.
Answer: The calibration curve has been added as Figure 5.
Remark: 6. The author should provide the related equation for calculating the LOD.
Answer: The detection limit was estimated not from an equation but from three times the standard deviation for the lowest studied Ga(III) concentration.
Remark: 7. The Stability of measurement section should come after Analytical performance section.
Answer: The order of the chapters Stability of measurement and Analytical performance has been changed accordingly.
Yours sincerely,
Prof. Malgorzata Grabarczyk
Round 2
Reviewer 1 Report
The authors do not clearly explain the increase in the signal.
“As proven in previous works, it is possible to concentrate the analyte using the solid bismuth electrode [21,2224]. In the proposed measuring system, a gallium peak was also observed after its accumulation as a result of reduction to a metallic form and then a signal was obtained after its oxidation. However, a very favorable significant increase was observed in the sensitivity of the determinations due to the activation of the solid bismuth electrode by introducing Bi(III) ions into the tested solution and their simultaneous reduction with the accumulated gallium. Activation of the electrode is obtained as a result of the precipitation of freshly reduced metallic bismuth on the surface of the solid bismuth electrode and due to that the electrochemical activity of the electrode increases”
This answer leaves many questions that should be answered by the authors before being published.
1.- What is the difference between the Bismuth obtained from the activation and the Bismuth from the solid electrode, which increases the current response?
2.- If the Bi film is formed ex situ and then the Bi sample is measured, does the current increase?
3.- Could a Ga-Bi intermetallic compound be forming?
4.- Could the electroactive area of the electrode be increasing?
5.- Are there any changes in the morphology that can interpret this increase in current?
